# Perceived Challenges to Anglophone Publication at Three Universities in Chile

**Marna Broekhoff**

American English Institute, Emerita, University of Oregon, Eugene, OR 97403, USA; marnab@uoregon.edu

**Abstract:** It is well documented that non-Anglophone scholars face enormous pressures to publish in peer-reviewed English-medium journals both for their own advancement and for institutional prestige. Yet many of them receive little support and face big challenges. Scholars' perceptions of these challenges often differ from those of applied linguists. This study analyzes publication challenges at three universities in Chile. Research questions included the following: How much pressure to publish do Chilean scholars feel? What do they perceive as their biggest challenges? Do these differ from perceptions of applied linguists? Data come from surveys involving 191 respondents conducted shortly after the author was an English Language Specialist for the U.S. Department of State in 2015. Surveys were administered both as personal interviews and online through SurveyMonkey. Identified challenges include language issues, workload, feedback and networking, and rhetorical structure. Suggestions are given for mitigating these challenges and for further research on these issues.

**Keywords:** networking; rhetorical structure; Anglophone; periphery; bilingual; academic writing

## 1. Introduction

The pressure on global scholars to publish in English, particularly in internationally indexed English-medium journals, has been well documented [1–4]. Indeed, that was the theme of the PRISEAL (Publishing and Presenting Research Internationally for Speakers of English as an Additional Language) IV International Conference in Reykjavik in September 2018 as well as previous PRISEAL conferences. English is now commonly considered the hegemonic language of publication not only in science, technology, engineering, and mathematics (STEM) fields but also in humanities and social sciences [4,5]. Indexed databases abound for both, including PubMed, EBSCO, SCI, and Scopus in the former, and SSCI and A&HCI in the latter. Publishing in English, regardless of topic or desired audience, is widely seen as essential both for individual advancement and for institutional prestige [4,6] (pp. 1, 8). Many Ph.D. programs now require publication in indexed journals as a pre-requisite for graduation [7]. These publishing requirements have been described as "a high-stakes game upon which hiring, promotion, and continued employment can depend" [8] (p. 3).

Despite such pressure, English as an additional language (EAL) scholars outside the Anglophone center often receive little support, either from their professions or from their institutions, to get published. Insufficient professional support and other issues have led to a new acronym, ERPP (English for Research Publication Purposes) introduced in a special issue of the *Journal of English for Academic Purposes* [9]. Responding to the same problems two decades earlier, Swales [10] used social-constructivist theory to introduce the concept of discourse communities, or professionals with common goals, participatory mechanisms, information exchange, community-specific genres, highly specialized terminology, and a high general level of expertise. To join a discourse community and thus receive professional support for disseminating one's own research, a scholar must learn the conventions that underlie these six criteria. The concept of discourse community has been further developed by Casanave [11]. Flowerdew [2] also explains the difficulties in joining them for scholars in the Anglophone periphery.

The pressure on scholars to publish and the need for more support to do so were the motivating factors for the Universidad Magallanes (UM) in Punta Arenas, Chile, to partner with the U.S. Department of State, Office of English Language Programs, to hire me as an "English Language Specialist" to create an academic writing center at their institution in 2015. Although the State Department also wanted encouragement for students to study in the U.S., the university itself was more interested in support for professors. Therefore, my primary task was to find ways to help professors get published and to mentor them through this process. During this assignment I became aware that most of my clients wanted to focus primarily on linguistic issues that plague most EAL writers, whereas I thought meta-linguistic issues of structure and thesis were also and perhaps more problematic, along with issues of networking in this remote location near the Straits of Magellan at the southern tip of the country. I found that my concerns were echoed in the work of publication researchers.

This discrepancy between what my professorial clients said they wanted and what publication researchers say they need motivated me and several colleagues to design a survey questionnaire near the end of my five-month tenure in Chile and after my return to the U.S. Our study was designed to answer the following questions from a larger sample than those who had visited the new writing center during my time there:

(1)　How much pressure and desire do Chilean scholars feel to publish?
(2)　What have been their experiences with publishing or attempting to do so?
(3)　What do they perceive as their biggest challenges to publishing?
(4)　Do their perceptions differ from those of writing center directors and other applied linguists?

Thus, the rationale for this study was to try to identify Chilean researchers' concerns more clearly and to resolve the apparent disconnect between their views and those of applied linguists.

## 2. Literature Review

Most EAL scholars themselves think that linguistic issues of grammar, syntax, and lexis are the highest hurdles for seeing their research in print [12–15]. This concern with language is not surprising when one reads critical and often patronizing comments by journal editors and other "gatekeepers". These mostly focus on linguistic deficiencies, reinforcing the attitude that meaning and grammar in the English language are arcane and unchanging entities. Curry and Lillis reprint sample comments from editors that are often quoted: "There are quite a number of instances in the text where it is clear that English is not the authors' first language. They cannot be blamed for that, of course, but it does make the text difficult to understand at times". "The author is clearly well-versed in this area, but the manuscript needs quite a bit of work on its English grammar and spelling. A native speaker should correct the manuscript" [13] (pp. 135,136). Hyland argues that "the frequency of these comments, and occasionally their bluntness, may lead EAL writers to believe that language has played a decisive role in the rejection of their contributions" [16]. Reflecting such comments 15 years earlier, Flowerdew claims that articles are often rejected on the basis of language competence alone [14]. Cameron has coined the term "verbal hygiene" to denote the irresistible urge to legislate language matters with style guides, etc., especially among publishing houses [17]. From her interviews with eight article reviewers, Strauss concludes that "it appears that there is a bias against language that differs from native speaker use, and that authors who employ nonstandard English might well be regarded negatively, regardless of the merits of their research" [18] (p. 1).

Opinions about the importance of language skills are far more nuanced among applied linguists, especially publication researchers, than among journal editors. Most argue that language skills are a necessary but not sufficient requirement for publication success. For example, Lillis and Curry have argued for the importance of feedback from "literacy brokers" perhaps through an editing service, as well as from academic, or "content brokers", such as senior professors, local and international colleagues, and editorial assistants from the target journal [19]. Others have gone further and argued that there is little evidence that matters of language and non-nativeness are clear factors in the rejection

or acceptance of a manuscript [13,20]. Hyland claims in a much-criticized article that the arcane language of research publication is no more difficult for EAL scholars than for native speakers [16], thus levelling the playing field regarding language for all speakers of English. Claiming that linguistic injustice is a myth, he argues instead that the quality of the research is more important than the language used to report it [16].

Of course one should acknowledge here that EAL writers do not comprise a homogeneous group; their challenges in writing English vary greatly with their exposures to it, their uses of it in their research settings, and their position on the trajectory from novice to expert in their fields. In her ethnographic study of "situated multilingualism" at an institution in Norway, Nygaard argues that progression on this trajectory is most important for getting published [21]. In their study of Spanish academics at a university in Spain, Ferguson, Perez-Llantada, and Plo argue that scholars' perception of language issues is "complex and multidimensional", depending more on self-reported language proficiencies than on the influence of their native language (in this case, Spanish) [20].

Instead of language, most publication researchers argue that networking is the primary determinant of publication success. Broadly defined, networking can encompass feedback from a colleague about one's article draft; negotiation with editors; collaboration with colleagues both local and abroad; attendance at conferences and follow-up communication; and access to information about conferences, grants, and publishing opportunities. In their ethnographic studies over more than a decade of 50 European scholars in four countries, Curry and Lillis have written extensively about the importance of networking throughout the research and publication process [1,6,13,19,22,23]. Other writers, including Li and Flowerdew [2,24,25], have done ethnographic studies of networking resources in China, such as supervisors, peers, and language professionals. Networking is obviously more important for novice researchers than for established scholars, but it is also more difficult for them [26].

Implicit in these comments about networking is the importance of funding, both institutionally and in the larger contexts of country and region. Hultgren claims that wealth is a key factor in a nation's likelihood to participate in global knowledge production. Obviously, gross inequities exist—only 10 countries produce more than half the world's academic output, while the remaining 221 countries produce the rest [27] (p. 8). She concludes that "the overriding factor in explaining this inequity appears to be, not whether or not English is a research writer's first language, but the resources and networks that are available to them" (p. 2). Curry agrees in a recent interview [26], and these all depend on funding. "To get your work published, it may be more important that you find yourself in the right environment . . . " concludes Hultgren (p. 9). Canagarajah's seminal work in 2002 and afterwards explains the many disadvantages of working in the "periphery" outside the Anglophone center both geographically and culturally, as well as economically [28].

Less attention is paid in the literature to rhetorical structure than to networking—or to language, although few would deny its importance in producing publishable work. Kaplan's classic study from the 1960s [29] (which has since been criticized) analyzes the striking difference in rhetorical patterns between Anglophone and Asian, Latin, and other cultures. He claims that in the dominant Western rhetorical orientation, the main point, or thesis, is expected in the opening paragraphs, with succeeding paragraphs presenting subpoints and supporting detail in a very linear progression. In other cultures, the writer approaches the subject from all sides in a circular or zigzag pattern and saves the thesis for the end, after the reader has "earned" it. More recently, Connor also comments on these pattern differences in her explanation of contrastive rhetoric [30]. Other research on this topic revolves around structural templates proposed by Swales starting in the 1990s, including his Introduction, Methods, Results, Discussion (IMRD) structure for the entire research article and his Create a Research Space (CaRS) model for Introductions [10,31]. These templates are used and taught widely. Their overall structure is in the shape of an hourglass, going from general to specific in the Introduction, and specific back to general in the Discussion. Within the all-important introduction, Swales argues that the author must create a research space (CaRS) through three identifiable "moves": Establishing a research context, establishing a niche in this context, and occupying the niche [31]. In his study of Taiwanese graduate

students, Huang argues for a Swalesian genre-based research writing course for all of them [7]. Morell and Cesteros describe the benefits of their bilingual (Spanish and English) course on genre awareness for increasing their students' research skills [32].

However, Swales' templates are now viewed more fluidly than formerly, particularly in social sciences and humanities. Curry argues that scholars should pay more attention to the structure of articles in their target journals than to rigid templates [33]. Addressing the use of templates in culturally and linguistically diverse research communities, Perez-Llantada proposes a "multiliterate rhetorical consciousness-raising pedagogy". In this bibliometric study, she defends generic variations and attributes them to the impact of the authors' native languages [34]. Similarly, in their examination of article introductions over a 20-year period, Solli and Odemark argue that multilingualism causes variations in generic structure [35].

From the apparent discrepancy between the issues in the literature identified above and the concerns of our writing center clients, our survey project was born.

## 3. Methods

### 3.1. Demographics

Data come from both interviews and online questionnaires administered to 191 survey respondents at three universities: The Universidad Magallanes at the southernmost tip of Chile has an enrollment of about 3000 with undergraduate and graduate programs focusing on biology, education, and law. Its professorial staff numbers 200, including adjuncts and part-timers. The Universidad de Talca (TA) just south of Santiago has an enrollment of about 11,000 with emphases on health, agriculture, business/economy, and law, and a faculty of 500. The Pontificia Universidad Catolica de Chile (PUC) in the Santiago capital, one of the largest and most prestigious institutions (often ranked third) in South America, provides 20 major programs in the sciences, engineering, medicine, and humanities, among others. It has an enrollment exceeding 30,000 and a professoriate of 3500. Half of those completing the survey at the Pontificia have Ph.D.s or post-docs and more than 10 years' experience. More than 40% of these respondents teach medicine or engineering. Further detail about these institutions and the 191 respondents' specific career stage, academic unit, education, and number of publications can be found in Table 1a,b below and in Appendices A.2 and B.2. Note that the column under Education gives percentages only for respondents, not the entire faculty.

**Table 1.** (**a**) Institutional demographics; (**b**) professorial demographics.

| (a) | | | |
|---|---|---|---|
| **Institution** | **# Students** | **Prestige** | **Academic Units** |
| UM (Punta Arenas) N = 26 | 3000 | Ranked 33 out of 60 in Chile | Biology Education Hard Sciences Law |
| TA (Talca) N = 5 | 11,000 | Ranked 16 out of 60 in Chile | Health Agriculture Business/Economics Law |
| PUC (Santiago) N = 160 | 30,000 | Ranked 2 out of 60 in Chile | Medicine Engineering Sciences Humanities |

**Table 1.** *Cont.*

| (b) | | | |
|---|---|---|---|
| **Institution** | **Faculty** | **Education** | **Years of Service** |
| UM<br>(Punta Arenas)<br>*N* = 26 | 200 | Ph.D.: 16<br>61% | >20 yrs: 3<br>~15 yrs: 4<br>~10 yrs: 2 |
| TA<br>(Talca)<br>*N* = 5 | 500 | Ph.D.: 4<br>80% | >20 yrs: 1<br>~15 yrs: 3<br>~10 yrs: 4 |
| PUC<br>(Santiago)<br>*N* = 160 | 3500 | Ph.D. or post-doc:<br>106 (67%) | >20 yrs: 85<br>~15 yrs: 42<br>~10 yrs: 33 |

### 3.2. Instruments

Designed to elicit respondents' attitudes about and experiences with publication, along with their perceived challenges during this process, the surveys included questions in three main areas with the last one receiving most attention in the surveys and responses because it is the most complex and the most amenable for help.

1. Scholars' pressure and desire for publication of their research, whether in English-medium international journals (EMI), English-medium regional journals (EMR), Spanish-medium international journals (SMI), or Spanish-medium regional journals (SMR);
2. Scholars' experiences with publication or publication attempts;
3. Their perceived challenges to getting published.

The questions were formulated by writing center staff at the three institutions, first written in English, then translated into Spanish by native speakers. The listed challenges were proposed by writing center directors based on their experiences in working with researchers. Opportunity was given for respondents to identify other challenges, but no one did. Due to time and funding constraints, they were not pilot-tested, which would have improved clarity of some items. Half the questions for the personal interviews were open-ended, whereas the other half for the interviews and most of those for the SurveyMonkey questionnaire were closed-ended, including multiple choice and rank ordering. See Appendices A.1 and B.1 for detailed lists of these questions. Surveys were administered in both English and Spanish in personal interviews as well as online through SurveyMonkey. Some replies were given in Spanish and some in English.

Regarding confidentiality for the data collection, although the names of respondents were known to the interviewers in the platform interviews, these names were removed before the data were submitted. The SurveyMonkey responses are by nature anonymous. Thus, in both formats anonymity and privacy were preserved, and there were no ethical issues. Because the methodology complied with all policies at the institution in question, further permission was not required. The named writing center directors who participated in data collection all saw earlier drafts of this article. A representative of the English Language Programs at the U.S. Embassy in Santiago, sponsor of my work in Chile, also reviewed this article.

Although administration of the surveys started out as personal interviews between writing center directors and their professorial clients at UM and TA, it soon became clear after two months with only 31 responses at these two institutions that this approach was wholly inadequate. Besides the obvious time constraints of this approach, scholars are understandably sensitive about revealing their publication records and the impact of these on their careers. They may have been particularly circumspect in a face-to-face meeting with a writing center director they probably did not know well and who might report back to their department heads.

Subsequently a SurveyMonkey questionnaire was sent out to more than 2000 professorial staff at the PUC in Santiago. The 160 responses to the SurveyMonkey questionnaire were far more than the 31 interview responses from the other two institutions, but those 160 were nevertheless a small percentage of the total population that received the questionnaire. One can argue that the study needs to be replicated and expanded in hopes of getting a higher response rate, but achieving this is unlikely, given the sensitivity of the subject and the "bother" of responding to an impersonal computerized questionnaire. Considering this conundrum, rather than seeking statistics, it may be more revealing and productive to do ethnographic and longitudinal studies of individual scholars' publication efforts, as several researchers have done, most notably Curry and Lillis, and Li and Flowerdew [12,19,24,25,36–38].

*3.3. Analysis*

Tabulations of our results are detailed in Appendices A.2 and B.2 and in the tables below. Because our interest was in determining the needs of faculty clients of our writing centers, regardless of department, we did not try to correlate responses with departmental affiliation. Large writing centers at well-funded large institutions may have the luxury of focusing on discipline-specific needs and approaches, but new writing centers in developing countries (where I have created three) are unlikely to have the staff size or expertise to do so. Because the responses from UM and TA were few in number and spoken to a human interviewer, we did not see the value in reporting these separately in Figure 1 from the much larger sample from the SurveyMonkey questionnaire at the PUC. Despite some differences, we thought that reporting responses from all three universities in aggregate would give a clearer idea about perceptions and needs of Chilean scholars than would be the case with just one institution. Significantly, rankings and commentary from the personal interviews and the SurveyMonkey questionnaires mostly support and reinforce each other, with some exceptions duly noted in the Discussion.

The methodology used for analyzing the responses was simple: Tabulations for closed-ended responses at the two smaller universities were done by hand from interviewer notations on each questionnaire, and percentages were calculated. Responses to the open-ended questions were summarized and consolidated when similar. Numerical tabulations and percentages for each of the SurveyMonkey questions at the PUC were built into the software. Open-ended responses were summarized. Numerical tabulations from all three universities for the rank-ordering of publication challenges were then added together and percentages were calculated for the bar graph in Figure 1, although they can be viewed separately in the tables below and in the appendices, and they are treated separately in the Discussion section. Total responses for each question do not always add up to the total (31 or 160) because some respondents skipped some of the questions.

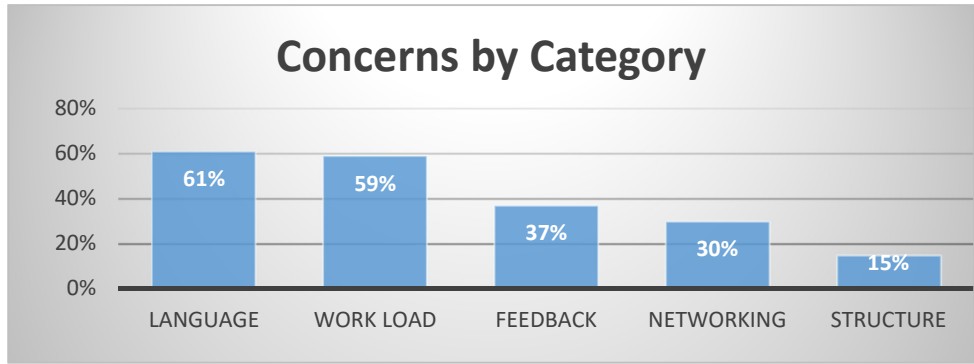

**Figure 1.** Combined survey responses at three Chilean universities rank-ordering perceptions of difficulty by percentages of total responses.

## 4. Results and Discussion

*4.1. Scholars' Desires and Experiences with Publication (Questions 1 and 2)*

Table 2 below shows that at all three universities, regarding the first two sets of questions, the vast majority of scholars want to publish in English. Their very legitimate reasons are that English is the language of science, it reaches the widest possible audience, and provides the greatest professional prestige. About three-quarters of respondents have achieved this goal with at least one publication. Percentages in all categories are consistently higher for the PUC. One researcher at TA commented that "English is the international language for science. Therefore, if you want to have your work read and cited, publish in English" (TA1). Perez-Llantada would certainly agree, with her statement that "English has become a lingua franca not because it has been imposed but … because it facilitates the exchange of scientific knowledge worldwide." [33] (p. 11). Others bemoan the dominance of English as "epistemicide" of alternative ways of knowing [39]. Most scholars are also keenly aware of the importance of publishing not just in English, but in indexed, high-impact journals in English. Certainly this lofty goal is elusive, even for native English speakers with superb writing skills, as the acceptance rate for perhaps the most prestigious international scientific journal, *Nature*, is only about 8% [40]. This low acceptance rate is an enormous problem with no easy solutions, guaranteed to cause professional frustration and less attention to other professional duties, such as teaching [5].

As further shown in Table 2, about half of respondents also want to publish or have published in their native language, usually Spanish, whether in an SMI or SMR journal, in addition to publishing in English. Their reasons include wanting to publish regionally relevant research for regional audiences who would have the most interest in it, such as social problems or governmental issues. Surely the women's strike in Southern Chile in 2018 would be better covered in Spanish or in both English and Spanish, as would the ice physics research of the Patagonia Institute. As one respondent from UM put it, "Most important is the circuit of academic contacts with similar interests" (UM1) Pressures to publish in English or in one's native language are complex, depending on which language is used in a country or institution for professional communication [35] or on academic specialty [22]. Perez-Llantada explains that in the humanities, especially in Mexico, Portugal, and Spain, there is strong use of the national language [33]. In our own work with Chilean scholars, we found that regardless of specialty, most of them communicate almost exclusively in Spanish, yet they prefer to publish in English.

**Table 2.** Respondents' publication desires and experiences by institution.

| Institution | Desire for English | Desire for Spanish | Published in English | Published in Spanish, Other |
|---|---|---|---|---|
| UM and TA<br>*N* = 31 | 24 (75%) | 3 (10%) | >5: 10 (33%)<br>3–4: 6<br>1–2: 8<br>0: 7<br>Total pub: 24 (77%) | >5: 10 (33%)<br>3–4: 2<br>1–2: 9<br>0: 11<br>Total: 21 (67%) |
| PUC<br>*N* = 160 | 159 (99%) | 114 (71%) | >5: 66 (41%)<br>3–4: 21 (13%)<br>1–2: 37 (23%)<br>0: 36 (23%)<br>Total pub:124 (78%) | [Breakdown N/A]<br>65 (75%)<br>Other langs:16 (10%)<br><br>Total pub:125 (78%) |

Despite clear reasons sometimes to use one's native language, most respondents are keenly, if uncomfortably, aware that publishing in indexed English-medium international journals will lead much more directly to professional advancement, a concern that is well documented in publication research [41]. Several researchers in Punta Arenas were studying ways to make salmon farming more profitable in the Magellan Straits. Such a topic would more appropriately be discussed in the local Spanish language to reach more local entrepreneurs who might work on the project, but these

researchers wanted my help to publish in English. Unfortunately, the old adage "publish or perish'" has become "publish in prestigious English-medium journals or disappear from the profession". Given the low acceptance rate from exclusive indexed journals, it is easy to argue that this requirement is not sustainable and has already spawned predatory journals and the neglect of other professional duties [5].

As this brief analysis has made clear, scholars in Chile want to publish their research—preferably in English. What is holding them back?

### 4.2. Perceived Challenges to Getting Published (Question 3)

Regarding the third category of questions, respondents' perceived challenges to getting published, the aggregate results from all three institutions are shown graphically in Figure 1 above. Table 3 provides breakdowns by institution. Percentages are based on the total number of answers to each question, not the overall number of respondents. Some respondents at each institution chose not to respond to all the questions. The analysis that follows will point out where these perceptions differ from those of publication researchers and other applied linguists.

**Table 3.** Perceived challenges by institution given as percentages of total responses to interviews (UM and TA) and to questionnaire (PUC).

| Institution | Language | Workload | Feedback | Networking | Structure |
|---|---|---|---|---|---|
| UM and TA Total $N = 31$ | 20 (64%) Response $N = 31$ | 22 (88%) Response $N = 25$ | 6 (22%) Response $N = 30$ | 11 (37%) Response $N = 30$ | 4 (16%) Response $N = 25$ |
| PUC Total $N = 160$ | 74 (59%) Response $N = 126$ | 72 (59%) Response $N = 122$ | 58 (39%) Response $N = 148$ | 39 (29%) Response $N = 135$ | 19 (15%) Response $N = 126$ |

### 4.2.1. Language Issues

Nearly two-thirds (94 = 62% of 157) of those responding about language concerns at the three institutions claim that their primary challenge for publication is difficulty with the English language, having rank-ordered it as first or second, a problem they think could be solved simply with a good English editing service. Relatively fewer checked this box at the PUC than at the other two smaller and less prestigious schools. One respondent at UM commented that "The key element in my case is the 'correction of English' phase of the process and particularly the grammatical and composition-construction of the article" (UM2). Another said, "I think that face-to-face sessions with the staff of the Academic Writing Center may also improve my English writing skills. They could help me solving problems in terms of the language as well as having a different angle on my research" (UM3). Chilean scholars would probably agree with these expressions of frustration with the "language barrier" from half-way around the world: "I regard the language barrier as the central problem for Norwegian researchers". "One year in England/USA—even as a street sweeper—would likely mean more to a scientific career than half a million crowns in the form of a research grant" [10] (p. 102).

As noted in the first paragraph of the Literature Review [13,14,16,17], most editors would probably agree with these respondents' perceptions that language skills will make or break their publication success, but many applied linguists would not. As noted in the rest of the Literature Review, they claim that such issues as quality of research, professional expertise, location in the Anglophone center vs. periphery, and especially networking, are all more important. Although there are disagreements, most would probably claim that language skills are necessary but not sufficient for seeing one's research in print.

Thus, we have the first major discrepancy between survey respondents and publication researchers. Most of the former seem unaware of all the other stumbling blocks on the road to publication success.

### 4.2.2. Workload

Along with language issues, nearly as many responses (59%) about workload cited it as their major hindrance to both conducting and publishing research, with substantially more complaints about this at the two smaller universities. Many professors at Chilean universities teach 20 or more class hours per week and/or have heavy administrative duties, with no formal release time for research or writing. This was particularly noticeable at UM, where one respondent commented, "Too many classes impact productivity. Last semester I did 32.5 hours [of] classes" (UM4). Other teachers in the survey, however, have much fewer teaching and administrative tasks, thus creating great disparities in workloads within the same university. A respondent at UM commented, "Yes, [workload impacts my writing time], I have a lot of other responsibilities as Chief Editor of *Magallania* journal and coordinator of Centro de Estudios del Hombre Austral, plus all the management of research projects, participation on committees, etc." (UM5). Several respondents complained that they are assigned hours for research, but not for writing, although the latter takes nearly as long as the former. "This is exactly the hard issue that impacts negatively the academic writing. I already have two directorships plus teaching, and to write a paper involves a big process, like revisions, resubmissions, etc., which are not counted in terms of hours in our academic commitment" (UM6). Another one commented, [Administrators should] "recognize time enough to write articles. At present, the bureaucrats do not consider time for this very important activity. The time required should be defined by the scientists (not by bureaucrats)" (UM3). However, given the hierarchical structure of all Chilean institutions, including universities, negotiating work redistribution or getting more release time for writing are surely delicate matters.

Excessive workloads are certainly not limited to Chile but seem to characterize academic assignments in many developing countries and elsewhere in South America. For example, one writing center director from a university in Bogota, Columbia, explained the lack of response to our survey: "I'm sorry that my colleagues have not answered your survey. They are all extremely busy with courses, placement tests, English tutoring, English in the disciplines, and more. Thus they have not time or interest in publishing in recent years" [42].

Despite these challenges posed by academic workloads to finding time for research and writing as identified by the vast majority of respondents, there is relatively little discussion of this issue in the literature compared with other topics. Speaking of scholars in Kenya, Mweru cites lack of time as foremost along with lack of funding and lack of encouragement from university administrations as elements most hindering scholars in their efforts to publish [43]. Significantly, Kenya is a developing country, as is Chile, featured in this study. Most publication research to date has focused on countries that are developed (to a greater or lesser degree). It is hardly necessary to keep one's ear to the ground to know that workloads are a growing concern in the academic world both in developed and developing countries, yet publication research does not focus on workload as a major issue.

Workload can therefore be seen as the second major discrepancy between survey respondents and publication researchers—the former think it is a major concern in trying to publish; the latter mention it less than other issues.

### 4.2.3. Feedback and Networking

Feedback and networking trail far behind (37% and 30%, respectively) language issues and workload in the responses about the greatest publication challenges, as shown in Figure 1 above. Interestingly, the relative importance attributed to these two factors at the smaller schools was the reverse of that attributed at the larger one. Nevertheless, there was some recognition of the importance and difficulty of networking for scholars at the Universidad Magallanes in Chile's remote southern tip. This institution is isolated from the rest of the country geographically by the Andes mountains, and as a result it is also isolated socially and culturally. One respondent from this region commented, "Here in Chile, I have a big problem getting in contact with co-authors as they often do not respond to my offers to discuss the scientific content of a manuscript" (UM7). Another professor at this same institution commented, "[It is] easier to publish during my Ph.D. in Germany, because there exists a relationship

between researchers and journal editors" (UM5), but such comments are few. One respondent asked for "workshops and guidance in how to get published", "translators", and "lobbyists for entry to high-impact magazines [sic] in English" (UM8).

Contrary to the opinions of applied linguists, the majority of respondents to this question do not recognize the importance of feedback and networking, so they do not pursue contacts as much as they could. However, there are such opportunities in Chile, even in the remote Patagonian region. At UM these include more than 30 affiliations with other institutions, and regular symposia about environmental issues in this delicate sub-Arctic region. My writing center colleague at the PUC has received funding from the U.S. Embassy in Santiago to attend the TESOL Convention several years in a row, thereby providing her with new ideas for helping academics get published. However, survey respondents did not identify such opportunities as important to their publication success.

Despite these perceptions in our survey responses, applied linguists have argued for the prime importance of networking, as already described in the Literature Review, perhaps most notably Curry and Lillis in their longitudinal ethnographic studies [6,22,37]. They argue for the importance of networks that are strong, local, and long-lasting in helping scholars to participate in transnational networks that support their publishing in English and in local languages [22]. In a broader context than "literacy brokers", "publication brokers" need to be multi-lingual, multi-cultural, familiar with research article requirements in language and structure, and familiar with all the steps in the publication process, preferably having published themselves.

Such brokers are already provided by some journals or online through a burgeoning cottage industry, which gains status and credibility from the work of Lunsford and Ede, who have long argued that all good writing is collaborative [44]. A quick Google search yields more than 25 links to fee-based services for all aspects of writing an article for publication. On their website the editors of this journal, MDPI, suggest using a professional English editing service or their own "Regular Edit" covering grammar, spelling, punctuation, and style—for an extra fee. Their "Specialist Edit" includes advice from "an expert in your field" [45]. On the one hand, such services provide legitimate and much-needed help; on the other hand, they can be susceptible to exploitation and plagiarism. As noted in the Literature Review, these various networking opportunities depend heavily on funding, both specific to the researcher, project, or institution, and more generally, regarding the socioeconomic level of the country and region as well as its position relative to the Anglophone center.

Thus, we have a third discrepancy between survey respondents and publication researchers regarding the importance of feedback and networking.

### 4.2.4. Structure

The last and least challenge to publication success—in the minds of those responding to this category—is rhetorical structure, comprising only 15% in Figure 1 and similar for all three institutions. Most survey respondents said that structure "is no problem at all". With respect to research articles, several clients of the writing center at UM commented that merely summarizing one's research ought to be sufficient, without necessarily supporting a main point (UM3). Two exceptions, however, were at the PUC, where one professor expressed the cultural differences in rhetorical patterns quite succinctly: "The structure and flow of writing in English is radically different from Spanish. Writing in English requires specific knowledge that you are not taught even in the doctorate in English speaking countries" (PUC1). Another researcher argued that norms for rhetorical structure seem to be culturally derived, with his comment that it is a big problem "not to know nor to internalize the academic standards of the communities of which the most prestigious publications in English are a part . . . " [especially when] "many times he may have done his doctorate in different communities, with other standards and situated in other traditions" (PUC2). Most respondents, on the other hand, had little to say on this topic and gave it the lowest ranking. A caveat here may be the difference between perceived importance and perceived difficulty, as one publication researcher has pointed out the significant

difference between EAL writers' perceptions of the importance of discourse-level features and their perception of the greater difficulty of linguistic issues [46] (p. 100.)

In any case, the genre of Anglophone research articles can certainly be taught, as many applied linguists advocate at even the graduate level, thereby underlining its importance [7,15,32] and probably with less difficulty than linguistic competence. Swalesian templates are taught widely, as explained in the Literature Review—but not at the universities surveyed in our study. Other mentors of researchers advocate studying the structure and linguistic features of articles in target journals, especially if one wishes to publish in Spanish or other native languages [32,34].

To conclude this reporting of results, the final discrepancy identified in this study is between publication researchers who argue that structure is important and can and should be taught vs. survey respondents who seem unaware of this issue. My presence at UM was supposed to be a step in the right direction.

## 5. Recommendations

Now that publication challenges for EAL scholars have been identified and analyzed from the perspectives of both the scholars themselves and of applied linguists, the all-important task is to identify resources that can help them get published. Suggestions below come from survey responses to open-ended questions as well as from publication researchers. They are applicable to scholars working in Chile and globally.

Survey respondents' concern with English language issues is certainly well founded, even though it is not the only challenge or even the most important one in the opinions of publication researchers. However, help is available. Assistance with linguistic issues is already provided through academic writing centers at the three universities in this study and at others as well. There is even a South American Writing Centers Association (SAWCA) that hosts regional conferences and promotes communication and problem-solving among writing center directors in South America. Numerous survey respondents commented about the value of writing centers in their publication efforts, as previously noted.

In addition to writing center tutors, publication mentors exist at various levels to help scholars get published. Some researchers regularly work with native English-speaking (NES) "literacy brokers", who may or may not also be discipline specialists during their writing phase. One scholar at Magallanes commented that "you have to have English-speaking co-authors in some journals" (UM5). Chilean scholars would benefit from more publicity about these resources.

They might also feel encouraged by the newly popular concept of "world Englishes" [47], which is slowly supplanting the rigid semiotic unity of "standard academic English". A broadening of perspective among journal editors would have the double advantage of allowing larger participation in academic publication while also disseminating important research both inside and outside the Anglophone center. Researchers in Chile can certainly advocate for "world Englishes", but changes from the top down are usually slow.

Along with their concerns about English, survey respondents also expressed desires to publish in their native Spanish, particularly for subjects of regional or local interest such as social problems, governmental issues, or environmental issues unique to Patagonia. Already responding to this need, all three universities in the survey offer bilingual services through their writing centers. Although it is hard to change entrenched biases at the institutional level that tie promotion to publishing in indexed English-medium journals, professors, editors, and applied linguists in Chile and elsewhere could try to create more market for regional dissemination of research in languages other than English. Equivalent texts in different languages for different communities could be accepted and encouraged. Such equivalent texts should not be considered "self-plagiarism" or "dual submission" of English-only texts [48,49].

Regarding the second major challenge, workloads have to be negotiated on the institutional level, which would reap the benefits of spreading out teaching and administrative duties more evenly,

thereby reducing resentment and allowing more professors to participate in research—and writing. Complaints from respondents about inequitable workloads and no release time for writing have already been noted. Unfortunately, however, the hierarchical structure of all Chilean institutions, including universities, makes negotiating work redistribution or getting more release time for writing unlikely to occur.

Regarding networking, options are available, even in the Patagonian region, and could be enhanced. Department heads can and do work with institutional administrators to encourage conference attendance, networking, collaborative research, and even writing itself. Of course, they could do more in these areas as well as encourage use of academic writing centers. As noted earlier, the Universidad Magallanes has affiliations with 30 institutions throughout the world. International students and visiting professors provide networking opportunities on all three Magallanes campuses in Punta Arenas, Puerto Williams, and Puerto Natales. In the less geographically remote Santiago region, the other two universities in this study have even more connections.

Lastly, regarding structure, the genre of Anglophone research articles can certainly be taught, and with less difficulty than linguistic competence, as one researcher has pointed out [44] (p. 100). Again, academic writing centers can play an important role here by helping researchers not just with language issues but also with critical thinking and the structure of various types of academic discourse. As initiator of the writing center at Magallanes, I presented several workshops on Swales' concept of rhetorical structure, abstracts, and cover letters. I also worked intensively on structure and thesis with individual clients. Later, with the help of the U.S. Embassy, the current writing center director solicited U.S. experts to give several multi-day workshops on various aspects of academic writing. In the opinions of those attending these events, such support was very helpful. However, its availability needs to be much more publicized throughout the university. As in many universities, the departments at Magallanes tend to be isolated, with little interaction. More access to this instruction would be guaranteed if it were required by department heads, at least for graduate students, as it is in Taiwan [7]. Another obvious suggestion for all writers is to study the structure and linguistic features of articles in target journals. Most writers I counseled already do that.

## 6. Conclusions

This study tabulated 191 responses at three universities in Chile, a high number but a low percentage response rate, as approximately 2000 people were contacted via email. One can argue that a higher response rate is needed to validate the results, but achieving this is unlikely, particularly through the impersonal medium of a computerized survey. This study also included 31 personal interviews of the total 191, but it would be hard to conduct interviews on any large scale due to time constraints. Moreover, scholars are understandably sensitive about revealing their publication records and the impact of these on their careers. Some applied linguists have responded to this dilemma by conducting intensive longitudinal studies of selected EAL scholars' experiences with publication [12,22,24,36,38], but the relatively small samples may not be representative of the whole.

Despite the statistical limitations of this study, it has answered the four research questions posed in the Introduction. First and second, it confirms the enormous pressure—and desire—felt by Chilean scholars to publish in English, a pressure they share with scholars around the world, which has been well documented [1–4]. It also places Chilean scholars well outside the Anglophone center, due partly to their geographic isolation, particularly in the remote south, thus making it difficult for them to find support in various kinds of discourse communities described by Swales [10], Flowerdew [2], and others.

Regarding the third research question, perceived challenges to publishing, this study has identified the major challenges for publishing in English that Chilean scholars, as indeed all scholars, whether Anglophone or multilingual, must face; namely, language, workload, feedback, networking, and rhetorical structure. Most related studies have focused on developed countries in Europe or Asia, such as Ge in mainland China [4], Flowerdew in Hong Kong [12], Cho in Korea [15], Gosden in

Japan [50], and Curry and Lillis in four European countries [22], whereas this one moves to a developing country in South America but confirms similar findings. Although some results of the current study may seem surprising, such as the heavy workloads at the Chilean institutions, other results are intuitive, such as the language issues.

Answers to the fourth question, comparing perceptions of scholars with those of applied linguists, have revealed striking differences between the two, which have been the focus of this study. Whereas most EAL scholars see linguistic issues as their highest hurdles [12–15], publication researchers also emphasize networking [1,2,6,13,22,24] and rhetorical structure [7,10,38,51,52]. My own experiences highlight many Chilean scholars' geographic and cultural challenges to attaining networking and feedback, their overwhelming teaching and administrative loads (at least from my North American perspective), and their unfamiliarity with the structure of Anglophone research articles. This study also recognizes, both implicitly and explicitly, that the emphasis on English hinders the dissemination of research everywhere, not to mention the researchers themselves, as survey respondents noted and many scholars increasingly acknowledge [1,5,6].

This disconnect between the views of respondents and applied linguists could surely be lessened and publication increased by various forms of assistance. Regarding language issues, assistance can include the following: "literacy brokers", promotion of "world Englishes" [47], not just "standard academic English", and regional publication of "equivalent texts" in local languages without accusations of "dual submission" or "self-plagiarism" [48,49]. Workload adjustments can be made more equitable, and release time provided for both research and writing. Regarding networking, help can come from the new enterprise of "publication brokers" [1,6,44] and promotion of conferences and collaboration opportunities. Regarding structure, there could be direct instruction in academic genres, partly through writing centers. Implementation of these suggestions at various levels, including international, national, institutional, and academic unit, could go a long way not only to help scholars get published but also to reduce the obsession with Anglophone publication in Chile and in academic cultures everywhere.

Statistical challenges notwithstanding, research on publication challenges should continue and expand because of the enormous and increasing pressure to publish that confronts all academics, both EAL and native speakers. Perhaps the "bother" of responding to a publication survey could be mitigated in a future study by providing some incentive to respond, such as a small payment (if budget allows), departmental recognition, or even extra release time to do research and writing. Future surveys of publication challenges could be further broken down into subcategories under linguistic, discourse, and rhetorical, and the types of assistance that might be available for each [46]. Further ethnographic case studies will also provide valuable information, especially about the process of publication from first submission of a draft to final appearance in print. Few studies have been done comparing EAL writers in developing vs. developed countries, as this one does, and these would probably reveal differences in the challenges they face. Future PRISEAL and other conferences will play an important role in disseminating this research.

**Funding:** This research received no external funding.

**Acknowledgments:** The author wishes to acknowledge assistance with survey data collection from Gracielle Pereira, head of the Academic Writing Center at Pontificia Universidad Catolica de Chile in Santiago; Rachel Jimenez-Lange, EducationUSA advisor and head of the Academic Writing Center at the University of Talca; and Christian Formoso, head of the Academic Writing Center at the Universidad Magallanes in Punta Arenas. Assistance was also provided by the U.S. Embassy in Santiago.

**Conflicts of Interest:** The author declares no conflict of interest.

## Appendix A

*Appendix A.1 Interview Questions about Publishing at Magallanes and Talca*

January 2017

1.   Personal information:
     Name
     Number of years of service
     Current position
     Graduate degrees, from where, in what subjects

2.   Have you published a research article (RA) in English?   Yes   No
     If Yes, how many?   1   2   3   4   5 or more

3.   Have you published a research article (RA) in Spanish?   Yes   No
     If Yes, how many?   1   2   3   4   5 or more

4.   Have you published a research article (RA) in another language?   Yes   No
     If Yes, how many?   1   2   3   4   5 or more

5.   Have you ever attempted to publish a RA? Describe the process you went through, number of rejections, reviewers' comments, etc.

6.   Have you attempted or published other academic writings, such as book reviews, conference proceedings, letters to the editor, etc.? Describe that process.

7.   Do you *want* to publish in English? Spanish? Please explain and specify the following: English-medium international (EMI), English-medium regional (EMR), Spanish-medium international (SMI) or Spanish-medium regional/local (SMR). Why do you want to publish in this type (or types) of journal?

8.   Rank-order the following challenges for you to publish in English, placing the biggest challenge first: Then comment on your ranking.

     a.   Problems managing heavy workload
     b.   Problems with English language
     c.   Problems with knowing how to structure the research article (RA)
     d.   Problems making contacts with possible collaborators, editors, conference organizers
     e.   Problems with getting feedback about my research and/or writing
     f.    Other?

9.   Describe your workload at the university. Does it impact your academic writing?

10.   How could the administration help you to get published?

11.   Are you familiar with the English Academic Writing Center? If not, how could it be publicized? If so, have you used it? What kind of help did you receive? How could the AWC help you more?

*Appendix A.2 Interview Tabulations for Magallanes and Talca*

January 2017 Researchers: Marna Broekhoff, Christian Formoso, Rachael Jimenez-Lange *N* = 31

**RESPONDENT PROFILES**
**Question (1b)** Years of professional service
         >20 years: 4
         ~15 years: 7

~10 years: 4
~5 years: 8
**(1c, e)** Academic Unit
Biological sciences: 8
Education: 7
Hard sciences/math: 4
Social sciences: 4
Health and Nursing: 4
Engineering: 2
Humanities: 2
**(1d)** Education:
Ph.D.: 20
Master's or Other: 11
**(2)–(4)** Number of publications (Several have published >20 articles)
English: >5 pubs: 10; 4 pubs: 3; 3 pubs: 2; 2 pubs: 6; 1 pub: 2; 0 pubs: 8
Spanish: >5 pubs: 10; 4 pubs: 2; 3 pubs: 0; 2 pubs: 5; 1 pub: 4; 0 pubs:11
Other languages: 4

**PUBLISHING PROCESS, REVIEWER COMMENTS** (open-ended)

**(5)–(6)** Many comments that research article submissions rejected because of English, more than Spanish. Ten comments about working with an NES for revision or translation. One comment about easier to publish in Germany than Chile due to contacts with editors.

**DESIRE TO PUBLISH–TYPE OF JOURNAL–REASONS**

**EMI:** English-medium international; **EMR:** English-medium regional
**SMI:** Spanish-medium international; **SMR:** Spanish-medium regional/local
**(7)** EMI: (23) because reach the broadest audience, provide the most professional prestige, required by institution, required to get research funding, English is the language of science
SMR: (3) because these "less risky" [for rejection] or more appropriate for Latin audience
EMR: (1) [4 did not respond]

**(8) PUBLICATION CHALLENGES RANK ORDERED** (0 = not considered a problem; see Table 1 in text)

English problems: #1: 12; #2:8; #3: 6; #4: 1; #5: 5; #0: 4
Work load: #1: 16; #2: 6; #3: 2; #4:1; #5:0; #0:0
Contacts: #1: 3; #2: 8; #3: 5; #4: 7; #5: 2; #0: 5
Feedback: #1: 2; #2: 4; #3: 9; #4: 1; #5: 6; #0:5
Structure for RA: #1: 2; #2: 2; #3: 2; #4: 11; #5: 3; #0:5

**(9) WORKLOAD** (open-ended)

Great disparity in workloads, from research only to research + 32 class hours per week. One job description is 70% administrative. Some researchers spend most time in the field, thus relieving them from administration. Many complaints that research writing must be done evenings and weekends.

**(10) ADMINISTRATIVE HELP NEEDED TO ENCOURAGE PUBLICATION** (open-ended)

Half (15) express need for help with the English language. Others need for reduction in administrative duties, formal release time for research and writing, fewer teaching hours, and help with networking; more incentives to publish through pay or promotions.

**(11) FAMILIARITY AND OPINIONS OF ACADEMIC WRITING CENTER** (open-ended) Need writing strategies, not just editing and revising, or translation services

Help with critical thinking. Visits to AWC should be part of regular workload

**Appendix B**

*Appendix B.1 SurveyMonkey Questions about Publishing at Pontificia University, Santiago*

February 2017

1. *Por cuanto tiempo ha estado involucrado con trabjo académico?* What is your number of years of academic service?

2. *A que Unidad Academica pertenece?* To which academic unit do you belong?

3. *Cual es su nivel educacional?* What is your highest level of education?

4. *Le interesa publicar algún articulo académico en Ingles? Espanol? Otro idioma?* Are you interested in publishing in English? Spanish? Other?

5. *Anteriomente, ha publicado algún articulo académico en Ingles? Espanol: Otro idioma?* Have you published a research article before?

6. *Solo si ha publicado en Ingles: Cuantos artículos?* How many articles have you published in English?

7. *Ha intentado alguna ves publicar o ha publicado algún otro escrito académico, por ejemplo, Critica/resena de un libro? Estructura de presentación? Carta al editor? Otro?* Have you attempted or published other types of academic writing?

8. *Si esta interesado en publicar: Le gustaria publicar en un medio?* In what types of journals would you like to publish?

9. *Des las siguientes opciones: Cuales representan de mejor manera sus razones para publicar en ciertos tipos de revistas especializadas? Seleccione todas las opciones que correspondan.* Why do you want to publish in those journals?

10. *En su opinión, Cual es el nivel de dificultad de las siguientes que comúnmente enfrentan los académicos chilenos para publicar en ingles, siendo 1 la mas difícil y 5 la menos difícil:* administrar la carga laboral; idioma Ingles; conocimiento acerca; establecer contacto; feedback Rank order the challenges for publishing in English, with the biggest first: workload, language, structure, networking, and feedback.

11. *Conoce o ha escuchado acerca del English UC Writing Center?* Are you familiar with the English UC Academic Writing Center?

12.  *Ha asistido a talleres grulpales en el English Academic Writing Center?* If so, have you attended workshops at the AWC?

13.  *Ha asistido a sesiones individuales en el English Academic Writing Center? Que aspectos de la escritura trabajo (ideas, organización, creación de borrador, edición de estructura, revisión)?* If so, have you attended individual sessions at the AWC? And which aspects did you work on (ideas, organization, structure, editing)?

*Appendix B.2 SurveyMonkey Tabulations for Pontificia Universidad, Santiago*

February 2017 *N* = 160

**RESPONDENT PROFILES**
**Question (1)** Years of professional service

>20 years: N/A
>10 years: 53% (85)
5–10 years: 26% (42)
<5 years: 21% (33)

**(2)** Academic Unit: 20 faculties

Medicine: 31% (49)
Engineering: 10% (16)
Others: <10%

Agriculture, Architecture, Art, Biology, Economics, Social Science, Communication, Literature, Law, Education, Philosophy, Physics, History, Math, Chemistry, Theology

**(3)** Education:

Post-doc: 21% (33)
Ph.D.: 46% (73)
Master's: 26% (41)
Licentiate: 8% (13)

**(5) and (6)** Number of publications

Engl.: Total: 78% (125) >5 pubs: 41% 4 pubs: 4%; 3 pubs: 10%; 2 pubs: 10%; 1 pub: 13%; 0:22%
Spanish: Total: 65% (104) Breakdown N/A
Other languages: 10% (16) esp. French, Portuguese

**(7)** Other academic publishing

Bk. Review: 22.5% (36); Conf. proceedings: 46% (74); Letter to editor: 23% (37); Other: 40%; (64)—mostly book chapters 30% of 64 (18)

**DESIRE TO PUBLISH–TYPE OF JOURNAL–REASONS**

**EMI**: English-medium international; **EMR**: English-medium regional
**SMI**: Spanish-medium international; **SMR**: Spanish-medium regional/local

**(4)** English: 99% (159); Spanish: 42.5% (68); Other: 7.5% (12) esp. French, Portuguese
**(8)** EMI: 94% (150) EMR: 25% (40)
**(9)** Because these most prestigious 88% (141); Most familiar with conventions and lang.; 11% (19)
**(4)** SMI: 41% (66) SMR: 30% (48)

**(9)** Because these are most prestigious in my area 17.5% (28); because English is one of the main difficulties and I prefer to write in my own language 10% (16) See other comments about Spanish 10%; (16) we have cultural debt to our region; these have more influence and application in Chile; Want regional people to read my work; Want appeal to Latin audiences

**(10) PUBLICATION CHALLENGES RANK ORDERED** (0 = not considered a problem; see Figure 1 and Table 3 in text)

English problems: #1: 41; #2:33; #3: 24; #4: 14; #5: 10; #0: 4
Work load: #1: 50; #2: 22; #3: 18; #4:13; #5:19; #0:0
Contacts: #1: 16; #2: 22; #3:32; #4: 32; #5: 28; #0: 5
Feedback: #1: 22; #2: 36; #3: 34; #4: 28; #5: 23; #0:5
Structure for RA: #1: 5; #2: 14; #3: 29; #4: 39; #5: 39; #0:0

**FAMILIARITY AND OPINIONS OF ACADEMIC WRITING CENTER** (Little awareness)

**(11)** Familiarity with AWC: Yes 16% (26); No 84% (134)
**(12)–(14)** Have used AWC workshops, other services: Yes 1% (2) No 99% (158)
Need writing strategies, not just editing and revising, or translation services
Help with critical thinking. Visits to AWC should be part of regular workload

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
