# Peer review of "Perceived Challenges to Anglophone Publication at Three Universities in Chile"

_publications, doi:10.3390/publications7040061_

Round 1
Reviewer 1 Report
General Comments:
I think the author has done an excellent job revising the original submission. Please extend my congratulation to the author. The manuscript is now ready to be accepted for publication with the proviso that the minor amendments listed below are taken care of:
Minor amendments:
1. Page 4, line 32: I think “… have PhD’s or post-docs…” should read ‘… have PhDs or post-docs…'
2. Page 4, Table 1, last column of the second row: The percentage should be added like for the other two universities. I know this will be very small given that only 4 out of 500 faculty have a PhD degree, but I think it should be added for consistency.
3. Page 6, line 25: “… the vast majority (90%) of scholars..” I am not clear where the 90% comes from. I’ve done a number of calculations based on the data provided in Table 2, but I’m still not sure. Am I missing anything? Please make sure the percentage is accurate.
4. Page 7, line 22: “Tables 3 provides…” should read ‘Table 3 provides…’
Reviewer 2 Report
This article offers an interesting insight into how Chilean academics view the challenges they face in their attempts to publish in accredited English medium journals. I feel with a little more work that this could be a very interesting article but at the moment it is, in places, a puzzling and frustrating read.
In the literature review, the writing, in part, is careless and this impacts on meaning. For example where the author deals with Nygaard’s arguments the word “latter” is used in such a way that the reader is not sure what it refers to. The first sentence of the next paragraph beginning with “Rather than language…” is so strangely configured that it needs to be read a few times before it makes sense. The last sentence in this paragraph notes that networking is “obviously more important” but neglects to say more important than what. In the last five lines of the paragraph preceding Methods the author appears to be making a number of diverse points but they are not coherently linked or explained which leaves the reader a little lost.
The Methods section is very confusing. The table is not much help either. It might be better to have two tables; the first giving demographic information about the three institutions, while the second gives information about the participants. I also found myself somewhat lost when reading the explanation about the data collection instruments. The responses from two of the universities came from interviews while at the third and largest participants answered a survey. I need more convincing that the interviews and the questionnaires could be consolidated so easily. It has always been my experience that a great deal more information and insight is gained in interviews than in questionnaires. 31 in-depth interviews should provide a good deal of data. I would like the author to expand on this point. I was also somewhat concerned when I read a list of the survey questions (p.15) to note that the interviewees were given what the authors had decided were the five greatest challenges they faced. There was no opportunity for them to disagree that these were the five greatest challenges. Is there an explanation for this? Was this selection a result of what was gleaned in the interviews? If so, it would be a good idea to make this explicit.
In the opening paragraph under Instruments the author notes that the last area received the most attention. It needs to be made clear who gave it the great most attention. I was also a little puzzled about the ethics involved. I understand the anonymity of the questionnaire but what about the interviews?
As far as the analysis is concerned is concerned I would also like a little more detail as to how the responses to the open-ended questions were analysed. Was the author looking for themes and patterns to emerge?
The table in the Results and Discussion section is confusing particularly the two columns headed Desire for English/Desire for Spanish. Does this mean these are the number of people who wanted to publish in English and Spanish? Should the other two columns not read published instead of publish? Just under this table there is a discussion of the adage “publish or perish” and the author says “many think this is simply not sustainable…” Who are the many? I found the discussion of this section a little superficial. The emphasis was on reporting the findings and not a great deal on an actual in-depth analysis of these findings. I wonder if this problem might not be addressed if the findings and discussion section were separated.
The concerns about excessive workloads, I think, needs to be better contextualised. Workloads are a growing concern in the academic world both in developed and developing countries.
I was very concerned about one sentence under Feedback and networking. The author writes “The Andes Mountains and social or cultural inbreeding make networking opportunities particularly hard to attain at the Univeridad Magallanes in Chile’s remote southern tip” (p.9). I don’t know what the author means by this but this sentence needs to be very carefully edited.
On page 10 mention is made of a “burgeoning cottage industry”. This is an interesting and important point and I think deserves a lot more detail.
The next section and dealt with structure. It is difficult to follow particularly the section “Another observed … other traditions”.
In the conclusions the author makes a number of suggestions as to what could be done to mitigate the challenges these writers face. I would like to have seen these better developed and the practical implications thought through more carefully.
As I said earlier, I believe that this article has potential but more work needs to be done.
Round 2
Reviewer 2 Report
Thank you for your detailed responses to the issues I raised. Most of my concerns have been addressed but there are just one or two places where I would like clarification.
The first issue has to do with the first couple of sentences under 3.1 which I think should read: From the issues in the literature identified above which are seemingly... has an enrolment of about 3000 with undergraduate and graduate programs focusing on biology, education and law.
The second issue has to do with Table 1b. I am afraid that I still find this table rather confusing. Wouldn’t it be a lot easier to simply have as headings: Institution, PhD or post doc; years of service.
In your response to the point I raised about excessive workloads I am not persuaded that very little has been written about this. The following are just a few articles that come up in a search:
Kenny, J., & Fluck, A. (2018).Research Workloads in Australian Universities. Australian Universities' Review, 60(2), 25-37.
Apaydin, C. (2012). The workload of faculty members: the example of educational faculties in Turkey. College Student Journal, 46 (1), 20.
Kramer, B., & Libhaber, E. (2016). Writing for publication: institutional support provides an enabling environment. BMC Medical Education, 16.
The last two articles are written about developing countries.
Finally, I believe that the paragraphs you have added about suggestions for help are very useful and I would like to see them included in the final article. I found these pages clear and easy to read, and would advise strongly against rewriting.
I do not wish to insist that these changes are made. I think it would improve the article but I leave this decision in the hands of the editors. There is only one point which I believe must be resolved before this article is published. That has to do with the sentence on the top of page 10 which talks about a "long history of inbreeding". My apologies here I should have made it clear that my objection is to the word inbreeding which has incredibly negative connotations. Are you suggesting that the academics in this region are inbred? What do you mean by this? I would be incredibly offended if I was told that I belonged to an institution that had in history of inbreeding.
Apart from this last issue I believe that the changes that have been made have improved the article.
